# Non-Equilibrium Protein Folding and Activation by ATP-Driven Chaperones

**DOI:** 10.3390/biom12060832

**Published:** 2022-06-15

**Authors:** Huafeng Xu

**Affiliations:** Roivant Sciences, New York, NY 10036, USA; huafeng.xu@roivant.com

**Keywords:** chaperones, Hsp70, Hsp90, non-equilibrium, protein folding

## Abstract

Recent experimental studies suggest that ATP-driven molecular chaperones can stabilize protein substrates in their native structures out of thermal equilibrium. The mechanism of such non-equilibrium protein folding is an open question. Based on available structural and biochemical evidence, I propose here a unifying principle that underlies the conversion of chemical energy from ATP hydrolysis to the conformational free energy associated with protein folding and activation. I demonstrate that non-equilibrium folding requires the chaperones to break at least one of four symmetry conditions. The Hsp70 and Hsp90 chaperones each break a different subset of these symmetries and thus they use different mechanisms for non-equilibrium protein folding. I derive an upper bound on the non-equilibrium elevation of the native concentration, which implies that non-equilibrium folding only occurs in slow-folding proteins that adopt an unstable intermediate conformation in binding to ATP-driven chaperones. Contrary to the long-held view of Anfinsen’s hypothesis that proteins fold to their conformational free energy minima, my results predict that some proteins may fold into thermodynamically unstable native structures with the assistance of ATP-driven chaperones, and that the native structures of some chaperone-dependent proteins may be shaped by their chaperone-mediated folding pathways.

## 1. Introduction

A commonly accepted view on protein folding is Anfinsen’s thermodynamic hypothesis [1]: the native structure of a protein is uniquely determined by its amino acid sequence, and it is the conformation of the lowest free energy. According to this view, a free energy gap separates the native structure and the denatured conformations, and protein folding is accompanied by a negative free energy change [2]. A protein, left to its own device and given sufficient time, will fold spontaneously to its native structure.

We now know that many proteins depend on the assistance of molecular chaperones for folding into their functional structures inside cells [3,4,5]. ATP-driven chaperones such as GroEL/GroES [6,7,8], Hsp70 [9,10], and Hsp90 [11,12,13,14,15,16,17] represent an important class of chaperones that consume chemical energy in their functions. Biochemical and structural studies have established that these chaperones undergo a cycle powered by ATP hydrolysis through open and closed conformations [10,18,19,20,21,22]. These chaperones can rescue their protein substrates from misfolded or aggregated structures and accelerate their refolding to their native structures [23,24,25,26,27]. This role of ATP-driven chaperones does not contradict Anfinsen’s thermodynamic hypothesis: proteins still fold into the most thermodynamically stable structures, but the chaperones enable them to do so within a physiologically reasonable time [28].

Recent experimental studies suggest that ATP-driven chaperones may play a thermodynamic role besides the kinetic one: they may stabilize proteins in their native structures out of thermal equilibrium, converting the chemical energy of ATP hydrolysis into the conformational free energy of their substrates [26,29]. Coincidental to these experimental studies, theoretical models were published around the same time that predicted such non-equilibrium stabilization [29,30,31]. In addition to quantitatively recapitulating the experimentally observed acceleration in folding kinetics, these models suggest that ATP-driven chaperones can maintain their protein substrates in their native structures at higher concentrations than thermodynamically permitted in the chaperone-free equilibrium. They explain why ATP hydrolysis is indispensable to the cellular functions of these chaperones, and in the case of Hsp70 [30] and Hsp90 [31], the critical roles of their respective cochaperones.

Here, I define non-equilibrium protein folding to be the phenomenon in which the native fraction of a protein is elevated by an energy-consuming process above its value in thermal equilibrium. Let $f_{N}=\left[N\right]/P_{0}$ be the steady state fraction of the protein substrate in its native structure in the presence of ATP-driven chaperones, and $f_{N,eq}={\left[N\right]}_{eq}/P_{0}$ be the native fraction in the chaperone-free equilibrium, where $P_{0}$ is the total protein concentration and [N] is the concentration of the protein in its native structure. Non-equilibrium protein folding occurs if $f_{N}>f_{N,eq}$, which of course requires energy consumption. I will introduce the gain factor of non-equilibrium folding
(1)$$g\equiv \frac{f_{N}}{f_{N,eq}}=\frac{\left[N\right]}{{\left[N\right]}_{eq}}$$
which measures the extent of out-of-equilibrium stabilization of the native structure. A protein that primarily occupies the non-native structures in equilibrium (i.e., $f_{N,eq}<0.5$) but its native structure in the presence of ATP-driven chaperones (i.e., $f_{N}>0.5$) would contest Anfinsen’s hypothesis.

Note that the native fraction in my definition of non-equilibrium folding includes only the free (i.e., not chaperone-bound) native protein because chaperones primarily bind to proteins that are at least partially unfolded [9,32]. There are, however, examples in which chaperone-bound proteins retain some native activity. For instance, glucocorticoid receptor (GR) can bind to its ligand when it is in complex with Hsp90 [33]. In this case, however, GR may still need to dissociate from Hsp90 to function as an active transcription factor. Thus, in this work, I will only consider non-equilibrium folding to a free, native protein.

Many mechanistic models have been proposed for chaperone-mediated protein folding [27,28,32,34,35,36]. One prevalent hypothesis regards the chaperones as unfoldases or holdases [37], in that their primary function is to rescue a misfolded or aggregated protein substrate and to hold it in an unfolded state. Upon release from the chaperones, the protein molecule has a certain probability of folding into its native structure [38]. Models based on this hypothesis provide an explanation of how ATP-driven chaperones accelerate the folding of the substrates to their inherently stable native structures, but they do not provide an explicit mechanism for the chaperones to transfer the chemical energy from ATP hydrolysis into the folding free energy of the substrate protein. It has been proposed that the ATP energy is used by the chaperones to achieve ultra-affinity in substrate binding [36].

It is often unclear whether a model will imply non-equilibrium protein folding (i.e., g>1), when microscopic reversibility [31,39] is rigorously enforced. Based on thermodynamic principles, I have previously established one requirement of non-equilibrium protein folding: the substrate protein must undergo a conformational change when it is bound to the chaperone [31]. Supported by biochemical and structural evidence [40,41,42], this is a key assumption in my models of chaperone-mediated protein folding that couple the conformational dynamics of the protein substrate with the ATP-driven, open-close cycle of the chaperones (Figure 1).

In this work, I introduce an additional requirement that an ATP-driven chaperone must satisfy to perform non-equilibrium protein folding. Specifically, I demonstrate mathematically that an ATP-driven chaperone must break at least one of the four kinetic symmetry conditions (Conditions 1–4 in Section 3.1) to use the energy from ATP hydrolysis for out-of-equilibrium stabilization of substrate proteins in their native structures. As discussed below, Hsp70, Hsp90, and GroEL/GroES each break a different subset of the symmetry conditions, thus they use different mechanisms to perform non-equilibrium folding. Despite the difference in their mechanistic details, I present a unifying principle by which symmetry breaking translates into non-equilibrium folding to the native structures.

In addition, I derive an upper bound on the extent to which an ATP-driven chaperone can elevate the native fraction of a substrate above its chaperone-free equilibrium value (Equation (Equation 56)). My results suggest that, for substantial non-equilibrium protein folding (i.e., g≫1) to occur, the chaperone—with the possible exception of chaperonins such as GroEL/GroES—must bind to an unstable intermediate conformation of the substrate, and the substrate protein must fold slowly on its own.

Whether Anfinsen’s hypothesis holds true for an individual protein can be experimentally tested by comparing the protein’s activity in the presence and in the absence of functional ATP-driven chaperones; I have previously proposed new experiments that may provide such tests on Hsp70- and Hsp90-mediated folding [30,31]. In this work, I propose a potential proteomics-level experiment that may help identify proteins that depend on ATP-driven chaperones for maintenance of their native structures.

My models of non-equilibrium protein folding imply that the native structures of some proteins may be shaped by the chaperone-mediated folding pathways. They raise the possibility of discovering natural proteins—and engineering novel proteins—that adopt different conformations in the presence and absence of the chaperones.

### Assumptions and Notations

To facilitate the exposition, I summarize the assumptions and notations in my model as follows:The substrate protein can convert among a set of conformations S, both when it is free in solution and when it is bound to the chaperone. I will use *M* to denote the misfolded/aggregated conformation and *N* the native conformation. In addition, I will consider two classes of intermediate conformations: the unfolded and misfold-tending (or aggregation-tending) conformation *U*, and the non-native but native-tending conformation *F*. To avoid a proliferation of symbols and to underscore the mechanistic commonality shared by protein folding and activation, in the discussion of kinase activation, I will use *M* to denote the inactive conformation, *N* the active conformation, *U* the inactive-tending conformation, and *F* the active-tending conformation.The chaperone can transition among a set of states I, each state *i* characterized by its conformational state (e.g., open or closed) and the numbers and the types (ATP vs ADP) of bound nucleotides.

My model includes the following reactions:The substrate in conformation *S* binds to the chaperone in state *i* with the association rate constant $k_{a,Si}$ and the dissociation rate constant $k_{d,Si}$:
(2)$$S+H_{i}\overset{k_{a,Si}}{\underset{k_{d,Si}}{⇌}}SH_{i}$$The free substrate in solution converts between conformation *S* and conformation $S^{\prime }$:
(3)$$S\overset{k_{S\to S^{\prime }}}{\underset{k_{S^{\prime }\to S}}{⇌}}S^{\prime }$$The corresponding conformational equilibrium constant is
(4)$$K_{SS^{\prime }}=\frac{k_{S\to S^{\prime }}}{k_{S^{\prime }\to S}}$$The substrate bound to the chaperone in state *i* converts between conformation *S* and conformation $S^{\prime }$:
(5)$$SH_{i}\overset{k_{S\to S^{\prime },i}}{\underset{k_{S^{\prime }\to S,i}}{⇌}}S^{\prime }H_{i}$$The chaperone transitions between state *i* and state *j* when it is bound to a substrate in conformation *S*:
(6)$$SH_{i}\overset{k_{S,i\to j}}{\underset{k_{S,j\to i}}{⇌}}SH_{j}$$

## 2. Materials and Methods

### 2.1. Proof That Symmetry Breaking Is Required for Non-Equilibrium Protein Folding

I will show that, under the symmetry conditions (Conditions 1–4 in Section 3.1), the steady state concentrations of the substrate satisfies, for any pair of conformations *S* and $S^{\prime }$,
(7)$$k_{S\to S^{\prime }}\left[S\right]=k_{S^{\prime }\to S}\left[S^{\prime }\right]\Leftrightarrow \frac{\left[S^{\prime }\right]}{\left[S\right]}=\frac{k_{S\to S^{\prime }}}{k_{S^{\prime }\to S}}=K_{SS^{\prime }}$$
where [S] (or $\left[S^{\prime }\right]$) is the concentration of the free substrate in conformation *S* (or $S^{\prime }$). Thus, the steady state ratio $\left[S^{\prime }\right]/\left[S\right]$ is unchanged from that in the chaperone-free equilibrium ${\left[S^{\prime }\right]}_{eq}/{\left[S\right]}_{eq}=K_{SS^{\prime }}$ for any pair of conformations *S* and $S^{\prime }$, including $\left[N\right]/\left[M\right]={\left[N\right]}_{eq}/{\left[M\right]}_{eq}$, and the chaperone is unable to increase the native concentration of the substrate above that in the equilibrium.

Letting $\left[H_{i}\right]$ be the concentration of the chaperone in state *i* and $\left[SH_{i}\right]$ the concentration of the substrate in conformation *S* bound to the chaperone in state *i*, the steady state condition for the reactions in Equations (Equation 2), (Equation 3), (Equation 5) and (Equation 6) is
(8)$$\begin{array}{ccc} 0=\frac{d[S]}{dt} & = & \sum_{i}\left(k_{d,Si}\left[SH_{i}\right]-k_{a,Si}\left[H_{i}\right]\left[S\right]\right)+\sum_{S^{\prime }}\left(k_{S^{\prime }\to S}\left[S^{\prime }\right]-k_{S\to S^{\prime }}\left[S\right]\right)\\ 0=\frac{d[SH_{i}]}{dt} & = & k_{a,Si}\left[H_{i}\right]\left[S\right]-k_{d,Si}\left[SH_{i}\right]+\sum_{j\neq i}\left(k_{S,j\to i}\left[SH_{j}\right]-k_{S,i\to j}\left[SH_{i}\right]\right)\\ & + & \sum_{S^{\prime }}\left(k_{S^{\prime }\to S,i}\left[S^{\prime }H_{i}\right]-k_{S\to S^{\prime },i}\left[SH_{i}\right]\right) \end{array}$$

According to Condition 1, the ratio $k_{a,S^{\prime }i}/k_{a,Si}$ does not depend on *i*. I denote this ratio as
(9)$$\frac{k_{a,S^{\prime }i}}{k_{a,Si}}=\gamma_{SS^{\prime }}K_{SS^{\prime }}^{-1}$$
where $\gamma_{SS^{\prime }}$ is a number that does not depend on *i*.

Consider a hypothetical, restricted system in which the substrate bound to the chaperone cannot change conformations (i.e., setting $k_{S\to S^{\prime },i}=0$ for all *i* and all pairs of *S* and $S^{\prime }$ in Equation (Equation 8)). Because the reaction $S⇌S^{\prime }$ is not part of any energy consuming cycle in this restricted system, $\left[S^{\prime }\right]/\left[S\right]=K_{SS^{\prime }}$ [31]. Let $\left\{\left[S\right]\right\}⋃\left\{\left[SH_{i}\right]|i\in I\right\}$ be the steady state concentrations of the substrate in conformation *S* in this restricted system, I will show that
(10)$$\begin{array}{ccc} \left[S^{\prime }\right] & = & K_{SS^{\prime }}\left[S\right] \end{array}$$
(11)$$\begin{array}{ccc} {}[S^{\prime }H_{i}] & = & \gamma_{SS^{\prime }}\left[SH_{i}\right] \end{array}$$
are the steady state concentrations of the substrate in conformation $S^{\prime }$, and that $\left[SH_{i}\right]$ and $\left[S^{\prime }H_{i}\right]$ satisfy
(12)$$k_{S\to S^{\prime },i}\left[SH_{i}\right]-k_{S^{\prime }\to S,i}\left[S^{\prime }H_{i}\right]=0\;\;\forall i\in I$$
for the original $k_{S\to S^{\prime },i}>0$ and $k_{S^{\prime }\to S,i}>0$. Thus, the steady state concentrations of the restricted system are also the solution to the original steady state condition in Equation (Equation 8), and Equation (Equation 7) holds (it is equivalent to Equation (Equation 10)).

To prove Equation (Equation 12), consider first an open state *i*. Thermodynamic cycle closure in the following reaction cycle (which does not consume chemical energy because the chaperone does not change state),
(13)$$\begin{array}{ccc} S & \overset{k_{S\to S^{\prime }}}{\underset{k_{S^{\prime }\to S}}{⇌}} & S^{\prime }\\ S^{\prime }+H_{i} & \overset{k_{a,S^{\prime }i}}{\underset{k_{d,S^{\prime }i}}{⇌}} & S^{\prime }H_{i}\\ S^{\prime }H_{i} & \overset{k_{S^{\prime }\to S,i}}{\underset{k_{S\to S^{\prime },i}}{⇌}} & SH_{i}\\ SH_{i} & \overset{k_{d,Si}}{\underset{k_{a,Si}}{⇌}} & S+H_{i} \end{array}$$
implies that
(14)$$\frac{k_{S^{\prime }\to S,i}}{k_{S\to S^{\prime },i}}\frac{k_{a,S^{\prime }i}}{k_{d,S^{\prime }i}}\frac{k_{d,Si}}{k_{a,Si}}K_{SS^{\prime }}=1$$
(15)$$\begin{array}{ccc} & \Leftrightarrow & k_{S\to S^{\prime },i}=k_{S^{\prime }\to S,i}K_{SS^{\prime }}\frac{k_{a,S^{\prime }i}}{k_{a,Si}}\frac{k_{d,Si}}{k_{d,S^{\prime }i}}=k_{S^{\prime }\to S,i}\gamma_{SS^{\prime }}\;\;(∵\;\mathrm{Equations}\;(9)\;\mathrm{and}\;(37))\\ & \Rightarrow & k_{S^{\prime }\to S,i}\left[S^{\prime }H_{i}\right]-k_{S\to S^{\prime },i}\left[SH_{i}\right]\\ & = & \left[SH_{i}\right]\left(k_{S^{\prime }\to S,i}\gamma_{SS^{\prime }}-k_{S\to S^{\prime },i}\right)\;\;(∵\;\mathrm{Equation}\;(11))\\ & = & 0 \end{array}$$

If *i* is a closed state such that $k_{a,Si}=k_{a,S^{\prime }i}=k_{d,Si}=k_{d,S^{\prime }i}=0$, Equation (Equation 14) no longer holds. According to Condition 4, however, the chaperone can reversibly transition between *i* and an open state *j* without the consumption of chemical energy, and, according to Condition 3, the transition rates between *i* and *j* do not depend on the conformational state of the bound substrate, i.e.,
(16)$$\frac{k_{S^{\prime },i\to j}}{k_{S,i\to j}}=\frac{k_{S^{\prime },j\to i}}{k_{S,j\to i}}=1$$

Thus, thermodynamic cycle closure in the following reversible reaction cycle
(17)$$\begin{array}{ccc} S & \overset{k_{S\to S^{\prime }}}{\underset{k_{S^{\prime }\to S}}{⇌}} & S^{\prime }\\ S^{\prime }+H_{j} & \overset{k_{a,S^{\prime }j}}{\underset{k_{d,S^{\prime }j}}{⇌}} & S^{\prime }H_{j}\\ S^{\prime }H_{j} & \overset{k_{S^{\prime },j\to i}}{\underset{k_{S^{\prime },i\to j}}{⇌}} & S^{\prime }H_{i}\\ S^{\prime }H_{i} & \overset{k_{S^{\prime }\to S,i}}{\underset{k_{S\to S^{\prime },i}}{⇌}} & SH_{i}\\ SH_{i} & \overset{k_{S,i\to j}}{\underset{k_{S,j\to i}}{⇌}} & SH_{j}\\ SH_{j} & \overset{k_{d,Sj}}{\underset{k_{a,Sj}}{⇌}} & S+H_{j} \end{array}$$
implies
(18)$$\frac{k_{S^{\prime }\to S,i}}{k_{S\to S^{\prime },i}}\frac{k_{S,i\to j}}{k_{S,j\to i}}\frac{k_{S^{\prime },j\to i}}{k_{S^{\prime },i\to j}}\frac{k_{a,S^{\prime }j}}{k_{d,S^{\prime }j}}\frac{k_{d,Sj}}{k_{a,Sj}}K_{SS^{\prime }}=1$$
(19)$$\begin{array}{ccc} & \Rightarrow & \frac{k_{S^{\prime }\to S,i}}{k_{S\to S^{\prime },i}}\frac{k_{a,S^{\prime }j}}{k_{a,Sj}}\frac{k_{d,Sj}}{k_{d,S^{\prime }j}}K_{SS^{\prime }}=1\;\;(∵\mathrm{Equation}\;(16))\\ & \Rightarrow & \frac{k_{S^{\prime }\to S,i}}{k_{S\to S^{\prime },i}}\gamma_{SS^{\prime }}=1\;\;(∵\mathrm{Equations}\;(9)\;\mathrm{and}\;(37))\\ & \Rightarrow & k_{S^{\prime }\to S,i}\left[S^{\prime }H_{i}\right]-k_{S\to S^{\prime },i}\left[SH_{i}\right]\\ & = & \left[SH_{i}\right]\left(k_{S^{\prime }\to S,i}\gamma_{SS^{\prime }}-k_{S\to S^{\prime },i}\right)\;\;(∵\mathrm{Equation}\;(11))\\ & = & 0 \end{array}$$

Thus, Equation (Equation 12) is true for both open and closed states.

To prove that $\left\{\left[S^{\prime }\right]\right\}⋃\left\{\left[S^{\prime }H_{i}\right]|i\in I\right\}$ in Equations (Equation 10) and (11) satisfy the steady state condition Equation (Equation 8) (swapping $S^{\prime }$ and *S*), I only need to show that, for the reactions in Equations (Equation 2) and (Equation 6), the flux in each reaction involving the substrate in conformation $S^{\prime }$ is $\gamma_{SS^{\prime }}$ times the flux of the corresponding reaction involving the substrate in conformation *S* because $\left\{\left[S\right]\right\}⋃\left\{\left[SH_{i}\right]|i\in I\right\}$ satisfies Equation (Equation 8) and the reactions in Equations (Equation 3) and (Equation 5) have zero flux (Equations (Equation 10) and (Equation 12)).

Let $J_{S,ij}=k_{S,i\to j}\left[SH_{i}\right]-k_{S,j\to i}\left[SH_{j}\right]$ be the reactive flux of the state transition for the chaperone bound to a substrate in conformation *S* (Equation (Equation 6)) and $J_{Si}^{a}=k_{a,Si}\left[H_{i}\right]\left[S\right]-k_{d,Si}\left[SH_{i}\right]$ be the reactive flux of the substrate in conformation *S* binding to the chaperone in state *i* (Equation (Equation 2)). The corresponding reactive fluxes for the substrate in conformation $S^{\prime }$ are
(20)$$\begin{array}{ccc} J_{S^{\prime },ij} & = & k_{S^{\prime },i\to j}\left[S^{\prime }H_{i}\right]-k_{S^{\prime },j\to i}\left[S^{\prime }H_{j}\right]\\ & = & \gamma_{SS^{\prime }}\left(k_{S,i\to j}\left[SH_{i}\right]-k_{S,j\to i}\left[SH_{j}\right]\right)\;\;\left(∵\mathrm{Equation}\;(11)\;\mathrm{and}\;\mathrm{Condition}\;3\right)\\ & = & \gamma_{SS^{\prime }}J_{S,ij} \end{array}$$
and
(21)$$\begin{array}{ccc} J_{S^{\prime }i}^{a} & = & k_{a,S^{\prime }i}\left[H_{i}\right]\left[S^{\prime }\right]-k_{d,S^{\prime }i}\left[S^{\prime }H_{i}\right]\\ & = & k_{a,S^{\prime }i}\left[H_{i}\right]K_{SS^{\prime }}\left[S\right]-k_{d,Si}\gamma_{SS^{\prime }}\left[SH_{i}\right]\;\;(∵\;\mathrm{Equations}\;(10),\;\;(11)\;\mathrm{and}\;(37))\\ & = & \gamma_{SS^{\prime }}\left(k_{a,Si}\left[H_{i}\right]\left[S\right]-k_{d,Si}\left[SH_{i}\right]\right)\;\;(∵\mathrm{Equation}\;(9))\\ & = & \gamma_{SS^{\prime }}J_{Si}^{a} \end{array}$$

Q.E.D.

### 2.2. Derivation of the Upper Bound of the Native Concentration at the Steady State of Non-Equilibrium Folding

To derive the upper bound in Equation (Equation 51), consider the reactions in Table 1. These are simplifications of the reactions in Equations (Equation 2), (Equation 3), (Equation 5) and (Equation 6): only a substrate in intermediate conformations S=U,F can bind to the chaperone (see Section 3.2.1), and only two chaperone states, open (*O*) and closed (*C*), are considered. The results hold as long as the substrate binds to all chaperone open states with the same association and dissociation rate constants, i.e.,
(22)$$\begin{array}{ccc} k_{a,Si} & = & k_{a,S}\\ k_{d,Si} & = & k_{d,S}\;\;\forall \;\mathrm{open}\;\mathrm{state}\;i \end{array}$$

Let
(23)$$J_{FU}=k_{F\to U}\left[F\right]-k_{U\to F}\left[U\right]$$
be the reactive flux from *F* to *U*. At the steady state, there is no net flux into or out of any molecular species, implying
(24)$$\begin{array}{ccc} J_{FU} & = & k_{a,U}\left[U\right]\left[O\right]-k_{d,U}\left[UO\right]\\ & = & k_{d,F}\left[FO\right]-k_{a,F}\left[F\right]\left[O\right] \end{array}$$

Because no external chemical energy is consumed in the reaction cycle of
(25)U+O⇌UO⇌FO⇌F+O⇌U+O,
we have
(26)$$\frac{k_{F\to U,O}}{k_{U\to F,O}}\cdot \frac{k_{a,F}}{k_{d,F}}\cdot \frac{k_{d,U}}{k_{a,U}}\cdot \frac{k_{U\to F}}{k_{F\to U}}=1$$

Thus,
(27)$$\begin{array}{ccc} \frac{k_{F\to U,O}\left[FO\right]}{k_{U\to F,O}\left[UO\right]} & = & \frac{k_{F\to U,O}}{k_{U\to F,O}}\cdot \frac{k_{a,F}}{k_{d,F}}\cdot \frac{k_{d,U}}{k_{a,U}}\cdot \frac{k_{U\to F}}{k_{F\to U}}\\ & \cdot & \frac{k_{d,F}\left[FO\right]}{k_{a,F}\left[F\right]\left[O\right]}\cdot \frac{k_{a,U}\left[U\right]\left[O\right]}{k_{d,U}\left[UO\right]}\cdot \frac{k_{F\to U}\left[F\right]}{k_{U\to F}\left[U\right]}\\ & = & \frac{k_{d,F}\left[FO\right]}{k_{d,F}\left[F\right]\left[O\right]}\cdot \frac{k_{a,U}\left[U\right]\left[O\right]}{k_{d,U}\left[UO\right]}\cdot \frac{k_{F\to U}\left[F\right]}{k_{U\to F}\left[U\right]} \end{array}$$

If $J_{FU}$ in Equations (Equation 23) and (Equation 24) is positive, all three ratios on the right-hand side of Equation (Equation 27) are greater than 1; if $J_{FU}<0$, they are all smaller than 1. Thus, the reactive flux
(28)$$J_{UF,O}=k_{U\to F,O}\left[UO\right]-k_{F\to U,O}\left[FO\right]$$
must be of the opposite sign of $J_{FU}$.

If the chaperone drives the substrate toward the native structure, we have $\left[F\right]/\left[U\right]>K_{F}=k_{U\to F}/k_{F\to U}$, implying $J_{FU}>0$ and $J_{UF,O}<0$. Because the flux from conformation *F* to *U* in free substrates must balance the total flux from conformation *U* to *F* in chaperone-bound substrates, the steady state reactive flux of the reaction UC⇌FC
(29)$$J_{UF,C}=k_{U\to F,C}\left[UC\right]-k_{F\to U,C}\left[FC\right]$$
satisfies
(30)$$J_{FU}=J_{UF,C}+J_{UF,O}<J_{UF,C}$$

Thus,
(31)$$\begin{array}{ccc} & & k_{F\to U}\left[F\right]-k_{U\to F}\left[U\right]<k_{U\to F,C}\left[UC\right]-k_{F\to U,C}\left[FC\right]\\ & \Rightarrow & k_{F\to U}\left[F\right]<k_{U\to F}\left(\left[U\right]+\frac{k_{U\to F,C}}{k_{U\to F}}\left[UC\right]\right)\equiv k_{U\to F}\left(\left[U\right]+\alpha \left[UC\right]\right)\\ & \Rightarrow & \left[F\right]<K_{F}\cdot \max\left(1,\alpha \right)\cdot \left(\left[U\right]+\left[UC\right]\right) \end{array}$$

We also have, per Equations (Equation 23) and (Equation 24),
(32)$$\begin{array}{ccc} & & J_{FU}=k_{F\to U}\left[F\right]-k_{U\to F}\left[U\right]=k_{a,U}\left[O\right]\left[U\right]-k_{d,U}\left[UO\right]<k_{a,U}\left[O\right]\left[U\right]\\ & \Rightarrow & \left(k_{U\to F}+k_{a,U}\left[O\right]\right)\left[U\right]>k_{F\to U}\left[F\right] \end{array}$$

At the steady state, there is no net flux in M⇌U or in F⇌N, thus
(33)$$\begin{array}{ccc} \left[M\right] & = & K_{M}\left[U\right]\\ \left[N\right] & = & K_{N}\left[F\right] \end{array}$$

Because
(34)$$\left[M\right]+\left[U\right]+\left[UC\right]+\left[F\right]+\left[N\right]<P_{0}$$
we have
(35)$$\begin{array}{ccc} P_{0} & > & (K_{F}^{-1}\max{\left(1,\alpha \right)}^{-1}+1)\left[F\right]+\left[M\right]+\left[N\right]\;\;(∵\;\mathrm{Equation}\;(31))\;\\ & = & \left(K_{F}^{-1}\max{\left(1,\alpha \right)}^{-1}+1\right)\left[F\right]+K_{M}\left[U\right]+K_{N}\left[F\right]\\ & > & \left(K_{F}^{-1}\max{\left(1,\alpha \right)}^{-1}+1\right)\left[F\right]+K_{M}\frac{k_{F\to U}}{k_{U\to F}+k_{a,U}\left[O\right]}\left[F\right]+k_{N}\left[F\right]\;\;(∵\;\mathrm{Equation}\;(32)) \end{array}$$

Thus,
(36)$$\left[F\right]<{\left(K_{M}K_{F}^{-1}{\left(1+\frac{k_{a,U}\left[O\right]}{k_{U\to F}}\right)}^{-1}+K_{F}^{-1}\max{\left(1,\alpha \right)}^{-1}+1+K_{N}\right)}^{-1}P_{0}$$
and plugging in Equation (Equation 33) yields the upper bound in Equation (Equation 51).

## 3. Results

### 3.1. Non-Equilibrium Folding Requires Kinetic Symmetry Breaking

I present a set of four symmetry conditions that, if all satisfied, forbids an ATP-driven chaperone from elevating the native concentration [N] of its substrate above the chaperone-free equilibrium concentration ${\left[N\right]}_{eq}$. A chaperone must break at least one of these symmetry conditions to be able to convert chemical energy into non-equilibrium stabilization of the native structure of the substrate. As I discuss below, different chaperones break different symmetry conditions, corresponding to different mechanisms of non-equilibrium protein folding and activation. The symmetry conditions are as follows:The ratio of association rate constants $k_{a,S^{\prime }i}/k_{a,Si}$ does not depend on the chaperone state *i* for all pair of substrate conformations *S* and $S^{\prime }$ and for all open state *i*.The dissociation rate constant $k_{d,Si}$ does not depend on the substrate conformation *S*, i.e.,
(37)$$k_{d,Si}=k_{d,i}$$
for all open state *i* and for all conformation *S*.The transition rates between chaperone states are independent of the conformation of the bound substrate, i.e., $k_{S,i\to j}$ does not depend on *S* for all pair (i,j).For every closed state *i* of the chaperone, there is an open state *j*, such that the chaperone can reversibly transition between states *j* and *i* without consuming chemical energy.

In Section 2.1 of Materials and Methods, I prove that, if these four symmetry conditions are all satisfied, the ratio between the concentrations of the free substrate in any two conformations—say, *S* and $S^{\prime }$—at the chaperone-mediated steady state is unchanged from that in the chaperone-free equilibrium, i.e., $\left[S^{\prime }\right]/\left[S\right]={\left[S^{\prime }\right]}_{eq}/{\left[S\right]}_{eq}$, which implies $\left[N\right]/\left[M\right]={\left[N\right]}_{eq}/{\left[M\right]}_{eq}$. Because chaperone-binding reduces the total concentration of the free substrate, the native concentration of the free substrate will be lower in the presence of chaperones than in the absence of chaperones, i.e., g<1. (As noted in the Introduction, I only consider non-equilibrium folding to a free native protein.)

The above results regarding symmetry conditions hold for an arbitrary number of substrate conformations. For simplicity, I will assume only four representative conformations in the substrate, S={M≡misfolded,U≡misfold-tending,F≡native-tending,N≡native}, in the following discussion.

#### 3.1.1. Requisites for Breaking the Binding and Unbinding Symmetries (Conditions 1 and 2)

The binding symmetry, Condition 1, is trivially satisfied if there is only one open chaperone state to which the substrate binds, or if the substrate binding rate does not depend on the chaperone state, i.e., $k_{a,Si}=k_{a,S}$. Note that the substrate in different conformations *S* may bind to the chaperone at different rates $k_{a,S}$, e.g., the substrate in an unfolded structure may bind to the chaperone faster than the substrate in a near-native structure, which is a common assumption in models of chaperone-mediated folding [34], but this conformation-selective binding alone does not permit non-equilibrium folding (defined by g>1).

Condition 1 is approximately satisfied if the substrate in different conformations and the chaperone in different states bind using the same interface. In this case, the association rate constant is approximately
(38)$$k_{a,Si}=p_{S}\times f_{i}\times k_{a}$$
where $p_{S}$ is the probability that the binding surface on the substrate becomes accessible in conformation *S*, $f_{i}$ is the probability that the binding surface on the chaperone is accessible in state *i*, and $k_{a}$ is the intrinsic binding rate between the two binding surfaces once exposed (Equation (Equation 38) assumes that the conformational fluctuations exposing and occluding the binding surfaces are fast compared to the overall binding). The ratio
(39)$$\frac{k_{a,S^{\prime }i}}{k_{a,Si}}=\frac{p_{S^{\prime }}}{p_{S}}$$
thus satisfies Condition 1.

Condition 1 is violated if the substrate binds to different binding surfaces on the chaperone depending on both the substrate conformation and the chaperone (open) state. This requires that the chaperone possesses multiple open states in which different binding surfaces are exposed. There has not been experimental demonstration of any ATP-driven chaperone breaking this symmetry condition.

The unbinding symmetry, Condition 2, is approximately satisfied if the chaperone binds to the substrate in different conformations using the same binding interface. The symmetry is broken if the substrate in different conformations form different protein–protein interactions with the chaperone.

In one limit of such binding interface change, the substrate in the misfold-tending conformation *U* with a slow dissociation rate $k_{d,U}$ may bind to the open chaperone and, after the chaperone closes, change to the native-tending conformation *F* in which its chaperone-binding surface is lost, so that, when the chaperone opens again after the ATP-driven cycle, the substrate unbinds rapidly with a fast dissociation rate $k_{d,F}\gg k_{d,U}$. This may happen in chaperones that can retain a substrate without a contact interface while allowing the bound substrate to change conformation from *U* to *F*. Hsp90 and GroEL/ES are two such examples: Hsp90 clamps its client kinase between its closed homo dimer with a central hole that may accommodate substantial conformational changes in the client [31,42], and GroEL/ES holds the substrate in its cavity, inside which the substrate may fold [43]. These two chaperones may break Condition 2 by this mechanism and thus perform non-equilibrium protein folding.

Cochaperones that simultaneously bind to the chaperone and to the misfold-tending, but not the native-tending, conformation of the substrate may help break Condition 2. When the substrate in the misfold-tending conformation is bound to the cochaperone, the substrate–cochaperone complex together has an extended chaperone-binding surface with contributions from both the substrate and the cochaperone, which decreases the substrate’s dissociation rate from the chaperone. Binding to and unbinding from the cochaperone, a substrate in the misfold-tending conformation has, in effect, a slower dissociation rate than the substrate in the native-tending conformation. One case in point may be that of Cdc37-assisted kinase activation by Hsp90, as discussed in the following.

#### 3.1.2. Cdc37 Enables Hsp90 to Differentiate between the Active-Tending and Inactive-Tending Conformations of a Client Kinase

Cdc37 is a cochaperone that specializes in assisting Hsp90 to activate client kinases [32,44,45]. Experimental evidence suggests that Cdc37 binds to a locally unfolded conformation of the client kinase [46], and that Cdc37 can simultaneously bind to a client kinase and Hsp90 [42,47,48]. Based on the cryo-EM structure of the Hsp90-kinase-Cdc37 complex [42] (Figure 2A), I have previously proposed a simple mechanism for Cdc37 to distinguish between the inactive-tending (*U*) and active-tending (*F*) kinase conformations, binding to the former with higher affinity than to the latter: in the inactive-tending conformation, the disordered DFG-loop of the kinase does not interfere with Cdc37 binding, whereas, in the active-tending conformation, the DFG-loop may be ordered into a configuration that results in steric clashes with Cdc37 [31] (Figure 2B,C). Thus, Cdc37 can help Hsp90 retain an inactive-tending client more than an active-tending client, and the effective rate of dissociation from Hsp90 is higher for a client in the active-tending conformation than for a client in the inactive-tending conformation (Figure 2D,E), breaking symmetry Condition 2.

This mechanism implies the following reaction path of Hsp90-mediated kinase activation:(40)$$\begin{array}{ccc} \; & \; & U\overset{+\mathrm{Cdc}37}{⇌}U\cdot \mathrm{Cdc}37\overset{+\mathrm{Hsp}90_{\mathrm{open}}}{⇌}\mathrm{Hsp}90_{\mathrm{open}}\cdot U\cdot \mathrm{Cdc}37⇌\mathrm{Hsp}90_{\mathrm{closed}}\cdot U\cdot \mathrm{Cdc}37\\ \; & \; & \overset{-\mathrm{Cdc}37}{⇌}\mathrm{Hsp}90_{\mathrm{closed}}\cdot U⇌\mathrm{Hsp}90_{\mathrm{closed}}\cdot F\overset{\mathrm{ATP}\to \mathrm{ADP}+\mathrm{Pi}}{⇌}\mathrm{Hsp}90_{\mathrm{open}}\cdot F\overset{-\mathrm{Hsp}90_{\mathrm{open}}}{⇌}F \end{array}$$

Clearly, this mechanism requires that Cdc37 can dissociate from the Hsp90-kinase complex after Hsp90 closes. This requirement is indeed consistent with the observed structure of the Hsp90-kinase-Cdc37 complex: the closed Hsp90 clamps the client kinase between its N- and C-lobes to prevent the kinase from unbinding, but Cdc37 wraps around the exterior of Hsp90 so that it can disengage from the closed Hsp90 (Figure 2A).

Both the N-terminal domain (NTD) and the C-terminal domain (CTD) of Cdc37 bind to the partially unfolded kinase [49,50]. Individually, NTD and CTD bind to the kinase with low affinities [50] (on the order of 100 μM), but the bipartite interaction between the complete Cdc37 and the kinase results in sub-micromolar affinity. Based on the cryo-EM structure of the Hsp90-kinase-Cdc37 complex, the bipartite interaction may lead to the encirclement of a Hsp90 protomer by the kinase-Cdc37 binary complex, thus preventing the kinase from slipping off Hsp90 (Figure 2D). As discussed above, the NTD of Cdc37 may not bind to the active-tending conformation of the kinase. This not only substantially diminishes the affinity of Cdc37 to the kinase (CTD alone binds with over two-hundred-fold lower affinity), it also breaks the encirclement of the Hsp90 protomer by the Cdc37-kinase binary complex, potentially allowing the latter to dissociate rapidly from Hsp90 (Figure 2F), followed by the conversion of the kinase to the active conformation.

A puzzling observation is that Cdc37 binds to both the inactive B-Raf kinase and the active B-Raf mutant B-Raf${}^{V600E}$ (which has the valine at position 600 mutated to a glutamate) with similar affinities [51]: $K_{D}=1.0$
μM for the wild-type B-Raf and $K_{D}=0.4$
μM for the mutant B-Raf${}^{V600E}$ [50]. This can be explained by the above proposal that Cdc37 binds with high affinity to the inactive-tending conformation of the kinase but with comparatively negligible affinity to the other conformations. Consider the conformational equilibrium among the inactive (*M*), the inactive-tending (*U*), the active-tending (*F*), and the active (*N*) conformations:(41)$$M\overset{K_{M}^{-1}}{⇌}U\overset{K_{F}}{⇌}F\overset{K_{N}}{⇌}N$$

If Cdc37 binds to the inactive-tending conformation *U* with a conformation-specific dissociation constant $K_{D}^{*}$, the apparent experimentally measured dissociation constant of Cdc37 binding to the kinase is
(42)$$\begin{array}{ccc} K_{D} & = & \frac{\left[P][\mathrm{Cdc}37\right]}{\left[U\cdot \mathrm{Cdc}37\right]}\\ & = & \frac{\left[P\right]}{\left[U\right]}\cdot \frac{\left[U][\mathrm{Cdc}37\right]}{\left[U\cdot \mathrm{Cdc}37\right]}\\ & = & \frac{K_{M}K_{F}^{-1}+K_{F}^{-1}+1+K_{N}}{K_{F}^{-1}}\cdot K_{D}^{*} \end{array}$$
where *P* represents the kinase in any conformation.

The equilibrium active fraction, on the other hand, is
(43)$${\left[N\right]}_{eq}/P_{0}=\frac{K_{N}}{K_{M}K_{F}^{-1}+K_{F}^{-1}+1+K_{N}}$$

Thus, it is possible for the wild-type and the mutant kinase to have very different active fractions ${\left[N\right]}_{eq}/P_{0}$ yet similar $K_{D}$’s. For example, the hypothetical sets of equilibrium constants in Table 2 would be consistent with the observed Cdc37 affinities of the wild-type B-Raf and the V600E mutant and with the mechanistic hypothesis [52] that the mutation destabilizes the inactive and—less so—the inactive-tending conformation (thus decreasing $K_{M}$ and increasing $K_{F}$).

#### 3.1.3. Cochaperone Hsp40 Enables Differential ATP Hydrolysis by Hsp70 Bound to a Substrate in Different Conformations

Hsp70-mediated protein folding is an example of breaking symmetry Condition 3. The Hsp70 chaperones, such as the bacterial DnaK, adopts an open conformation when its nucleotide binding domain (NBD) is occupied by ATP. Upon ATP hydrolysis, Hsp70 changes to a closed conformation [53,54] (Figure 3A). By itself, Hsp70 has a low basal ATP hydrolysis rate, but the J domain from the Hsp40 cochaperones—also known as J proteins—can stimulate Hsp70 and drastically increase its ATP hydrolysis rate [55,56].

Both Hsp40 and Hsp70 bind to exposed hydrophobic sites on a substrate protein [57,58] (Figure 3A–C). Consequently, a substrate with multiple exposed hydrophobic sites may simultaneously bind to an Hsp70 and an Hsp40. This induces the proximity between the chaperone and the cochaperone, resulting in accelerated ATP hydrolysis in Hsp70 and its transition to the closed state. Because a substrate in the misfold-tending conformation often exposes more hydrophobic sites than a substrate in the native-tending conformation [59], an Hsp70 bound to the former is more likely to be stimulated by a nearby Hsp40 bound to the same substrate molecule than an Hsp70 bound to the latter. By recruiting Hsp40 to accelerate the ATP hydrolysis in Hsp70, a substrate in the misfold-tending conformation induces a higher rate of transition by Hsp70 from the open state to the closed state than a substrate in the native-tending conformation, i.e., $k_{U,\mathrm{open}\to \mathrm{closed}}>k_{F,\mathrm{open}\to \mathrm{closed}}$, breaking symmetry Condition 3 (Figure 3D,E).

As a result of this symmetry breaking, an Hsp70 bound to a substrate in the misfold-tending conformation is more likely to be closed than one bound to a substrate in the native-tending conformation. Thus, a substrate is on average more quickly released from the Hsp70 if it is in the native-tending conformation than if it is in the misfold-tending conformation. This difference biases the substrate toward the native conformation [30].

#### 3.1.4. Hsp70 and Hsp90 Perform Non-Equilibrium Folding by Preferentially Releasing Substrate Proteins in Native-Tending Conformations

The cochaperone Cdc37 helps break symmetry Condition 2 in Hsp90-mediated kinase activation. The cochaperone Hsp40 helps break symmetry Condition 3 in Hsp70-mediated protein folding. Despite breaking different symmetries, Hsp70 and Hsp90 share the same kinetic consequence: both chaperones release a bound substrate in the native-tending (*F*) conformation faster than a bound substrate in the misfold-tending (*U*) conformation.

To see how this kinetic asymmetry promotes the native concentration, consider first a system in which the symmetry conditions are satisfied (Figure 4A). A substrate in the *U* conformation binds to the chaperone faster than a substrate in the *F* conformation. As a result, the reactive flux through the ATP-driven cycle of a chaperone bound to a substrate in the *U* conformation is higher than that through the cycle of a chaperone bound to a substrate in the *F* conformation. However, kinetic symmetry ensures that, at the steady state, the flux of *U* binding to the chaperone is the same as the flux of *U* unbinding from the chaperone; the same holds true for *F* binding to and unbinding from the chaperone, and there is no net flux between *U* and *F*. Under the symmetry conditions, there are two parallel, independent chaperone cycles with respective reactive fluxes:(44)$$J_{S+\mathrm{Hsp}}=\left(S\to S\cdot \mathrm{Hsp}\to S\cdot \{\mathrm{states}\;\mathrm{of}\;\mathrm{Hsp}\cdots \}\to S\cdot \mathrm{Hsp}\to S\right)\;\;\mathrm{for}\;S=U,F,$$
and
(45)$$J_{U+\mathrm{Hsp}}>J_{F+\mathrm{Hsp}}$$

However, there is no net flux between *U* and *F*:(46)$$J_{UF}=k_{U\to F}\left[U\right]-k_{F\to U}\left[F\right]=0$$

Thus, the ratio between *F* and *U* is unchanged from the chaperone-free equilibrium:(47)$$\frac{\left[F\right]}{\left[U\right]}=\frac{k_{U\to F}}{k_{F\to U}}=K_{UF}=\frac{{\left[F\right]}_{eq}}{{\left[U\right]}_{eq}}$$

Symmetry breaking disrupts the independence between this pair of chaperone cycles. The release of a substrate in the *U* conformation from the chaperone is inhibited: in the case of Hsp90, Cdc37 helps the chaperone retain the bound client kinase; in the case of Hsp70, Hsp40-stimulated ATP hydrolysis and closure in Hsp70 diminish the reactive flux to re-open the chaperone (Figure 4B). This forces part of the reactive flux in $J_{U+\mathrm{HSP}}$ after the binding of the substrate to be diverted into the reactive flux of conformation conversion in the bound substrate:(48)$$J_{U\cdot \mathrm{Hsp}\to F\cdot \mathrm{Hsp}}>0$$

This in turn leads to a corresponding increase in the reactive flux of the chaperone’s release of the substrate in the *F* conformation, which increases [F] such that
(49)$$\frac{\left[F\right]}{\left[U\right]}>\frac{{\left[F\right]}_{eq}}{{\left[U\right]}_{eq}}$$

Thus, symmetry breaking biases the substrate toward the native-tending conformation and elevates the native concentration.

#### 3.1.5. The Potential Role of Sequential Hydrolyses of Multiple ATPs in the Chaperone Cycle

Breaking symmetry Condition 4 permits a net reactive flux along the following path that promotes the native-tending conformation *F* over the misfold-tending conformation *U*:(50)$$\begin{array}{ccc} & & U\overset{+{\mathrm{Hsp}}_{\mathrm{open}}}{\to }U\cdot {\mathrm{Hsp}}_{\mathrm{open}|\mathrm{closed}}\overset{\mathrm{ATP}\to \mathrm{ADP}+\mathrm{Pi}}{\to }U\cdot {\mathrm{Hsp}}_{\mathrm{closed}}\to F\cdot {\mathrm{Hsp}}_{\mathrm{closed}}\\ & & \overset{\mathrm{ATP}\to \mathrm{ADP}+\mathrm{Pi}}{\to }F\cdot {\mathrm{Hsp}}_{\mathrm{open}}\overset{-{\mathrm{Hsp}}_{\mathrm{open}}}{\to }F \end{array}$$

A non-zero net flux of $U\cdot {\mathrm{Hsp}}_{\mathrm{closed}}\to F\cdot {\mathrm{Hsp}}_{\mathrm{closed}}$ does not violate thermodynamic cycle closure in this case because the reaction cycle in Equation (Equation 17) is no longer reversible—ATP hydrolysis occurs and chemical energy is consumed in that cycle—and thus Equations (Equation 18) and (Equation 19) no longer hold.

To break symmetry Condition 4, at least one closed state of the chaperone must be separated from all the open states by ATP hydrolysis. This requires at least two ATP to be hydrolyzed sequentially—not synchronously—per chaperone cycle, and the substrate has to change conformation between two ATP hydrolyses. Examples include Hsp90 that hydrolyzes two ATP molecules sequentially in its cycle [60] and the group II chaperonins in eukaryotes—such as TRiC/CCT—that hydrolyzes up to eight ATPs sequentially [61,62]. The role of such sequential ATP hydrolysis—and the consequent symmetry breaking of Condition 4—in non-equilibrium protein folding is an open question.

### 3.2. An Upper Bound of Non-Equilibrium Protein Folding and Its Implications

Having established the symmetry breaking requirements for non-equilibrium folding, I now derive an upper bound on the folding capacity of an ATP-driven chaperone. The key result is
(51)$$\left[N\right]<\frac{K_{N}}{K_{M}K_{F}^{-1}{\left(1+\frac{k_{a,U}\left[O\right]}{k_{U\to F}}\right)}^{-1}+K_{F}^{-1}\max{\left(1,\alpha \right)}^{-1}+1+K_{N}}P_{0}$$
where [O] is the concentration of free chaperone in the open state, the equilibrium constants $K_{N}$, $K_{M}$, and $K_{F}$, the kinetic rate constants $k_{U\to F}$ and $k_{a,U}$, and their corresponding reactions are summarized in Table 1, and
(52)$$\alpha \equiv \frac{k_{U\to F,C}}{k_{U\to F}}$$
is an acceleration factor to indicate any potential rate change in conformation conversion when the substrate is bound to the closed chaperone. The proof of Equation (Equation 51) is given in Section 2.2 of Methods and Materials.

Equation (Equation 51) gives a general upper bound applicable to any ATP-driven chaperone. The folding capacity of a specific type of chaperone needs to be calculated by detailed models [30,31], but it cannot exceed that given by Equation (Equation 51). This result allows an analysis of the common key factors in non-equilibrium folding without considering the mechanistic details of specific chaperones.

Introducing a combined equilibrium constant for the reaction $U\overset{{\tilde{K}}_{F}}{⇌}\left(F+N\right)$
(53)$${\tilde{K}}_{F}\equiv \frac{{\left[F\right]}_{eq}+{\left[N\right]}_{eq}}{{\left[U\right]}_{eq}}=\left(1+K_{N}\right)K_{F}$$

The upper bound in Equation (Equation 51) can be written as
(54)$$\left[N\right]<\frac{1}{K_{M}{\tilde{K}}_{F}^{-1}{\left(1+\frac{k_{a,U}\left[O\right]}{k_{U\to F}}\right)}^{-1}+{\tilde{K}}_{F}^{-1}\max{\left(1,\alpha \right)}^{-1}+1}\cdot \frac{K_{N}}{1+K_{N}}P_{0}$$

Compare this to the native concentration in the chaperone-free equilibrium
(55)$$\begin{array}{ccc} {\left[N\right]}_{eq} & = & \frac{K_{N}}{K_{M}K_{F}^{-1}+K_{F}^{-1}+1+K_{N}}P_{0}\\ & = & \frac{1}{K_{M}{\tilde{K}}_{F}^{-1}+{\tilde{K}}_{F}^{-1}+1}\cdot \frac{K_{N}}{1+K_{N}}P_{0} \end{array}$$

The non-equilibrium gain factor is thus bounded by
(56)$$g=\frac{\left[N\right]}{{\left[N\right]}_{eq}}<\frac{K_{M}{\tilde{K}}_{F}^{-1}+{\tilde{K}}_{F}^{-1}+1}{K_{M}{\tilde{K}}_{F}^{-1}{\left(1+\frac{k_{a,U}\left[O\right]}{k_{U\to F}}\right)}^{-1}+{\tilde{K}}_{F}^{-1}\max{\left(1,\alpha \right)}^{-1}+1}$$

#### 3.2.1. Chaperones Bind to Unstable Intermediate Conformations of Substrates to Drive Non-Equilibrium Folding

An implication of Equation (Equation 56) is that ATP-driven chaperones must bind to an intermediate unfolded conformation (*U*) of the substrate, not to the misfolded conformation (*M*) itself, to perform non-equilibrium folding, unless the conformation conversion of a substrate is accelerated when bound to the chaperone (i.e., $k_{U\to F,C}>k_{U\to F}$ hence α>1). This can be demonstrated by contradiction. If the substrate does not have an intermediate misfold-tending conformation and the chaperone directly binds to the misfolded conformation, i.e., *M* and *U* are the same, Equation (Equation 51) reduces to (by setting $K_{M}=0$)
(57)$$\left[N\right]<{\left({\tilde{K}}_{F}^{-1}\max{\left(1,\alpha \right)}^{-1}+1\right)}^{-1}\cdot \frac{K_{N}}{1+K_{N}}P_{0}$$
and the upper bound of the non-equilibrium gain factor becomes
(58)$$g<\frac{{\tilde{K}}_{F}^{-1}+1}{{\tilde{K}}_{F}^{-1}\max{\left(1,\alpha \right)}^{-1}+1}$$

In the absence of a mechanism for the substrate to accelerate its conformation conversion when it is bound to the chaperone (α≤1), g≤1, the chaperone cannot elevate the native concentration.

To my knowledge, accelerated folding of protein substrates when bound to a chaperone has only been reported for the GroEL/GroES chaperonins [43,63,64,65,66,67]. In general, steric hindrance from the chaperone is more likely to impede rather than to accelerate conformation conversions in a bound substrate; this impedance was observed for the rhodanese protein trapped in GroEL/GroES by a single-molecule experiment [68]. For Hsp90 and Hsp70, there has not been any experimental demonstration that a substrate exhibits faster conformation conversions when bound to the chaperone than when free in the solution. This suggests that chaperones, with the potential exceptions of chaperonins, must bind to intermediate unfolded conformations of the substrate proteins to drive non-equilibrium protein folding.

Assuming α≤1, the upper bound on the non-equilibrium gain factor becomes
(59)$$g<g_{max}=\frac{K_{M}{\left({\tilde{K}}_{F}+1\right)}^{-1}+1}{K_{M}{\left({\tilde{K}}_{F}+1\right)}^{-1}{\left(1+\frac{k_{a,U}\left[O\right]}{k_{U\to F}}\right)}^{-1}+1}$$

For the gain factor to substantially exceed 1, the following must be true:(60)$$K_{M}{\left({\tilde{K}}_{F}+1\right)}^{-1}\gg 1\Rightarrow K_{M}\gg 1$$

Equation (Equation 60) implies that the intermediate conformation *U* to which the chaperone binds must be intrinsically unstable, and it will predominantly convert to the misfolded conformation *M* in the absence of the chaperone. This result is intuitive: if the chaperone binds to a dominant conformation of the substrate, it will trap a substantial fraction of the substrate and hinder its folding to the native structure. As a result, the chaperone will be unable to elevate the native concentration. The difficulty to observe the chaperone-binding conformations in biophysical experiments [69] attests to their transiency.

#### 3.2.2. Chaperones Stabilize the Native Structures of Slow-Folding Proteins

Non-equilibrium folding also requires, as implied by Equation (Equation 59) and g≫1,
(61)$$\frac{k_{a,U}\left[O\right]}{k_{U\to F}}\gg 1\Leftrightarrow k_{U\to F}\ll k_{a,U}\left[O\right]$$

Taken together, Equations (Equation 60) and (Equation 61) suggest that chaperones stabilize the native structures of slow-folding proteins. Assuming the binding rate constant to be on the order of $k_{a,U}∼10^{6}$ /M/s, the spontaneous (i.e., without chaperones) refolding rate of the protein, which is approximately $K_{M}^{-1}k_{U\to F}$, should be much slower than 1 /s to admit effective non-equilibrium folding by chaperones at a concentration of [O]∼1
μM.

#### 3.2.3. ATP-Driven Chaperones Buffer Destabilizing Mutations

About 18% of protein molecules in the cell harbor at least one missense mutation due to errors in translation [70]. In addition, proteins incur mutations due to germline and somatic gene polymorphism [71]. Given that about 30–40% of random substitutions disrupt protein functions [72,73], most probably by loss-of-folding [74,75], it is likely that many cellular protein molecules have compromised thermal stability and the native structures of some will not be the free energy minima. ATP-driven chaperones may buffer such destabilizing mutations [76,77] and maintain the native concentrations of these proteins by non-equilibrium folding [30].

The missense mutations may alter one or more of the transition rates and the equilibrium constants in protein folding dynamics: e.g., it may decrease the thermal stability of the protein by increasing $K_{M}$, decreasing $k_{U\to F}$ or increasing $k_{F\to U}$ (hence decreasing $K_{F}=k_{U\to F}/k_{F\to U}$), or decreasing $K_{N}$. Assuming α≤1 as discussed above, the maximum native concentration mediated by a chaperone is
(62)$$\begin{array}{ccc} {\left[N\right]}_{max} & = & g_{max}{\left[N\right]}_{eq}\\ \;\\ & = & \frac{1}{K_{M}{\left({\tilde{K}}_{F}+1\right)}^{-1}{\left(1+\frac{k_{a}\left[O\right]}{k_{U\to F}}\right)}^{-1}+1}\frac{1}{1+{\tilde{K}}_{F}^{-1}}\frac{K_{N}}{1+K_{N}}P_{0} \end{array}$$

Equation (Equation 62) suggests that the capacity of ATP-driven chaperones to buffer a destabilizing mutation depends on both the wild-type substrate’s folding kinetics and how the mutation alters the kinetic parameters (Figure 5). For instance, chaperones may be more effective in buffering mutations that slow down the transition from the misfold-tending conformation (*U*) to the native-tending conformation (*F*)—i.e., decreasing $k_{U\to F}$ by e.g., stabilizing the *U* conformation—than mutations that destabilize the native state by decreasing $K_{N}$. Such differential buffering may play a role in selecting tolerated genetic variations and shaping their consequences in human disease [78].

## 4. Discussion

Breaking the symmetry Conditions 1–4 is necessary but on its own is insufficient for non-equilibrium folding. g>1 often requires both substantial deviation from the symmetry conditions and other enabling kinetic conditions, as exemplified by Equation (Equation 56). Detailed mechanistic models [30,31] are needed to quantitatively predict the extent of non-equilibrium folding. Nonetheless, these symmetry conditions can help assess whether a proposed mechanism of chaperone function will imply non-equilibrium folding.

Unlike equilibrium protein folding, the native yield of non-equilibrium protein folding depends not only on the equilibrium constants but also on the kinetic parameters of the folding reactions and the chaperone cycle. The native concentration of a substrate may change in response to the modulation of the step-wise kinetics of the chaperone cycle [30,31,79] by cochaperones [80], by mutations [81,82] and post-translational modifications [83,84] in the chaperones, and by pharmacological molecules [85]. Such modulations may be used by the cell to regulate proteostasis. They may also offer therapeutic opportunities.

Given both the theoretical models and the experimental evidence suggesting that ATP-driven chaperones can stabilize the native or active structures of substrate proteins out of the thermal equilibrium, Anfinsen’s hypothesis does not *need* to be true for protein folding in cells. ATP-driven chaperones may not only kinetically accelerate the folding of proteins to thermodynamically stable native structures, but also actively fold some proteins to native structures that are thermodynamically unstable.

Most proteins have evolved to be marginally stable [86,87]. If ATP-driven chaperones can indeed buffer destabilizing mutations and maintain the native structures and functions of unstable mutants, as discussed in Section 3.2.3, it is then plausible that the native structures of some proteins may have become thermodynamically unstable as a consequence of this chaperone-buffered evolution. They may not stay folded on their own, but depend on the energy-consuming chaperones to maintain their native structures.

How many proteins in a cell take exception to Anfinsen’s hypothesis and depend on non-equilibrium folding by a particular ATP-driven chaperone? Emerging proteomics techniques may help answer this question. For example, cell lysates may be subject to proteolytic digestion [88] and the resulting products analyzed by mass spectrometry (MS), identifying proteins with permissible digestion sites, which approximately reflect their state of folding [89]. This proteolysis-MS assay may be repeated for lysates incubated with chaperone inhibitors [90,91] or chaperone agonists [85]. Proteins more susceptible to proteolysis in the presence of chaperone inhibitors—and less susceptible in the presence of chaperone agonists—are candidates that may depend on the chaperone for non-equilibrium folding to their native structures. The lysates should be incubated in the presence of protein synthesis inhibitors so that the analysis can isolate the chaperone’s effects on *maintaining* the native structures of its substrates from its effects on the folding of their nascent chains; the former demonstrates non-equilibrium stabilization of thermodynamically unfavorable native structures while the latter may be attributable to kinetic acceleration of protein folding. This analysis may be more applicable to GroEL/GroES and Hsp70 than to Hsp90 because the latter mediates the late-stage folding and activation of its substrates [33,92], which may not be associated with significant changes in the protein disorder detectable by the proteolysis-MS assay.

### Implications for Protein Native Structures and Their Folding Pathways

My model of non-equilibrium protein folding and activation suggests the tantalizing possibility that ATP-driven chaperones may play a role in shaping the native structures of some proteins. Consistent with a previous experimental demonstration that chaperones alter the folding pathway of a substrate protein [93], my model implies that an ATP-driven chaperone may bias a substrate protein to fold along pathways that expose few cochaperone binding sites during folding, with consequences for the resulting structures.

Consider two conformations *M* and *N* of a substrate protein, where *M* is the free energy minimum but associated with a folding pathway inhibited by the chaperone, and *N* has a higher free energy but can be reached through a folding pathway uninhibited by the chaperone (Figure 6). The protein will fold into its free energy minimum conformation *M* in the absence of the chaperone, but, if the chaperone-induced pathway bias is sufficiently strong, it will fold into the alternative conformation *N* in the presence of the chaperone. Note that *M* can be an ensemble of rapidly inter-converting conformations, such as in intrinsically disordered proteins (IDP) or intrinsically disordered protein regions [94,95,96,97,98]. Can ATP-driven chaperones fold some IDPs into well-ordered structures?

It has been proposed that some proteins may fold into native structures that are more kinetically accessible than conformations of the lowest free energy [99]. Indeed, experimental observations have been reported that synonymous codon substitutions result in conformational changes in the translated proteins, due to kinetic changes in the co-translational folding of the nascent chain on the ribosome [100,101,102,103]. These results are consistent with the idea that the native structures of some proteins may be determined by kinetics rather than thermodynamics. One implication is that the solution to the structure prediction problem for such proteins in cell may depend on the solution to the protein folding problem, and in-cell protein folding may be an active, energy-dependent process [104]. Predicting the cellular conformation of these proteins—in the presence of ATP-driven chaperones—may require the search for folding pathways that limit the exposures of cochaperone-binding, e.g., hydrophobic, sites.

## 5. Conclusions

In this work, I have proposed a theoretical framework to analyze non-equilibrium protein folding by ATP-driven chaperones. The symmetry breaking conditions may help determine whether a chaperone—by a proposed mechanism of action—can convert the energy from ATP hydrolysis to the out-of-equilibrium stabilization of the native structures of its substrate proteins. I have discussed how Hsp70 and Hsp90 may have broken different symmetries and how the symmetry breaking enables them to perform non-equilibrium protein folding and activation. My models predict that some proteins may fold to native structures that do not correspond to the free energy minima, and that their native structures may be shaped by the chaperone-mediated folding pathways. These predictions may be tested by experiments, some of which I have suggested above.

## Figures and Tables

**Figure 1 biomolecules-12-00832-f001:**
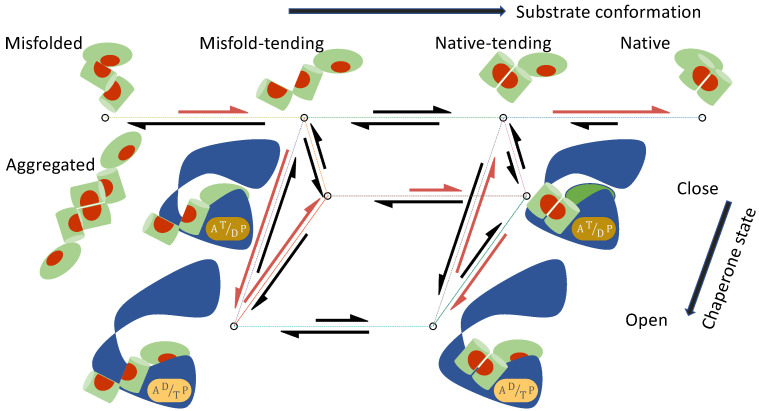
A mechanistic model of chaperone-mediated non-equilibrium protein folding that couples the state cycle of the chaperone and the conformational dynamics of its substrate. The chaperone undergoes a cycle of open and closed conformations, driven by ATP hydrolysis and nucleotide exchange. The protein substrate can transition among four classes of conformations: Misfolded (*M*), Misfold-tending (*U*), Native-tending (*F*), and Native (*N*). The lengths of the reaction arrows signify the corresponding reaction rates. The red arrows indicate the predominant folding pathway.

**Figure 2 biomolecules-12-00832-f002:**
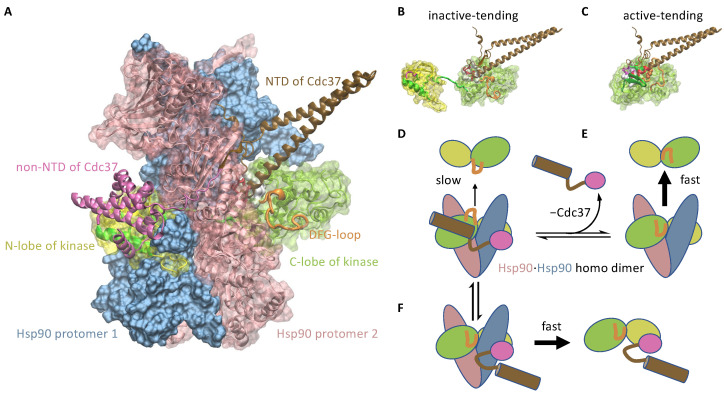
Cochaperone Cdc37 enables a client kinase in different conformations to unbind from Hsp90 at different rates. (**A**) the Hsp90-Cdk4-Cdc37 complex structure (PDB: 5FWM). The closed Hsp90 homo dimer clamps a partially unfolded Cdk4 kinase, and Cdc37 simultaneously binds to Cdk4 and Hsp90; (**B**) Cdc37 can bind to the kinase in the inactive-tending conformation; (**C**) steric clashes prevent Cdc37 from binding to the kinase in the active-tending conformation, due to its DFG-loop configuration and other conformational features; (**D**) Cdc37 helps to retain an inactive-tending kinase molecule inside the open Hsp90, resulting in slow unbinding of the kinase from the Hsp90. The bipartite interaction by NTD and CTD of Cdc37 with the kinase may result in the encirclement of a Hsp90 protomer by the Cdc37-kinase complex, preventing the latter from slipping off Hsp90. (**E**) Without Cdc37, an active-tending kinase molecule unbinds rapidly from the open Hsp90. (**F**) Alternatively, the loss of the interaction between the NTD of Cdc37 and the C-lobe of an active-tending kinase breaks the bipartite interaction between Cdc37 and the kinase, resulting in the release of the Cdc37-kinase complex from Hsp90.

**Figure 3 biomolecules-12-00832-f003:**
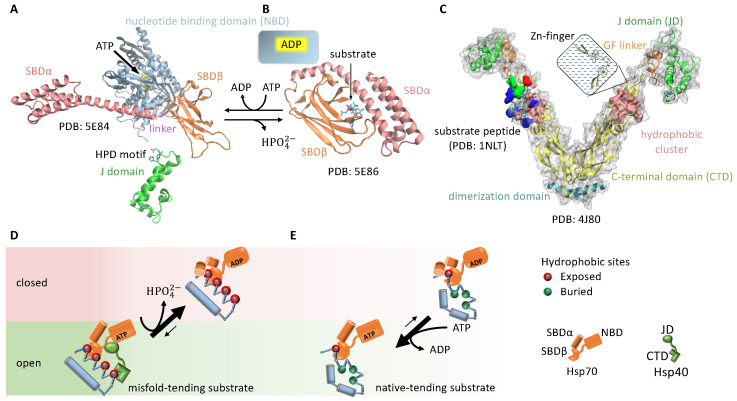
Cochaperone Hsp40 enables Hsp70 to change the balance between its open and closed states in response to the conformation of a bound substrate. (**A**) the ATP-bound, open state of Hsp70, which allows rapid binding and unbinding of the substrate; (**B**) the ADP-bound, closed state of Hsp70, with slow binding and unbinding of the substrate. SBD: substrate binding domain. (**C**) the structure of the Hsp40 cochaperone, including CTD that can bind to exposed hydrophobic sites on a substrate and the J domain that can stimulate the ATP hydrolysis of Hsp70. (**D**) An Hsp70 bound to a misfold-tending substrate molecule with many exposed hydrophobic sites is likely to be in proximity to an Hsp40 bound to the same substrate molecule, thus the Hsp70 will be stimulated in ATP hydrolysis, which drives the Hsp70 to its ADP-bound, closed state. (**E**) An Hsp70 bound to a native-tending substrate molecule with few exposed hydrophobic sites is unlikely to have a nearby Hsp40 and thus unlikely to be stimulated in ATP hydrolysis, and nucleotide exchange drives the Hsp70 toward its ATP-bound, open state.

**Figure 4 biomolecules-12-00832-f004:**
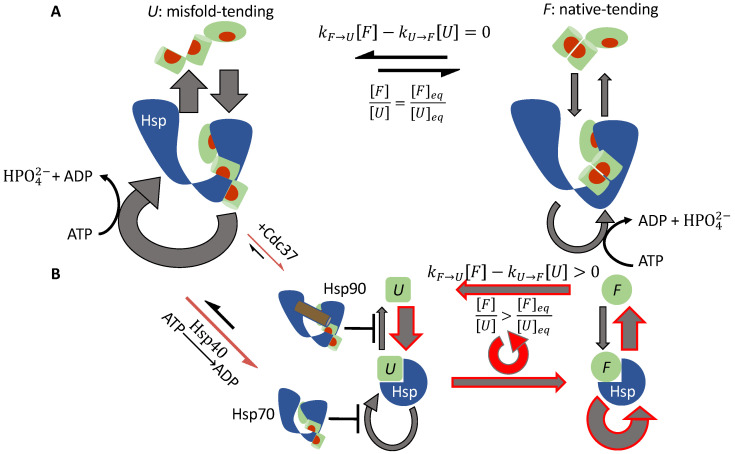
Reactive flux in chaperone-mediated non-equilibrium protein folding. (**A**) Under the symmetry conditions, there are two independent ATP-driven chaperone cycles: one with a higher reactive flux for a substrate in the misfold-tending (*U*) conformation (left) and one with a lower reactive flux for a substrate in the native-tending (*F*) conformation (right). There is no net flux between the substrate’s two conformations, and the ratio [F]/[U] is the same as its chaperone-free equilibrium value. (**B**) Cochaperones break the kinetic symmetry. The release of a substrate in the *U* conformation from the chaperone is inhibited: Cdc37 assists Hsp90 with retaining the substrate and Hsp40 stimulates ATP hydrolysis and closure of Hsp70. This restricts the reactive flux to release a substrate in the *U* conformation, forcing a partial diversion of the flux to the conformation conversion from U·Hsp to F·Hsp and resulting in a net reactive flux of U→U·Hsp→F·Hsp→F→U (red cycle), which elevates the ratio [F]/[U] above its chaperone-free equilibrium value.

**Figure 5 biomolecules-12-00832-f005:**
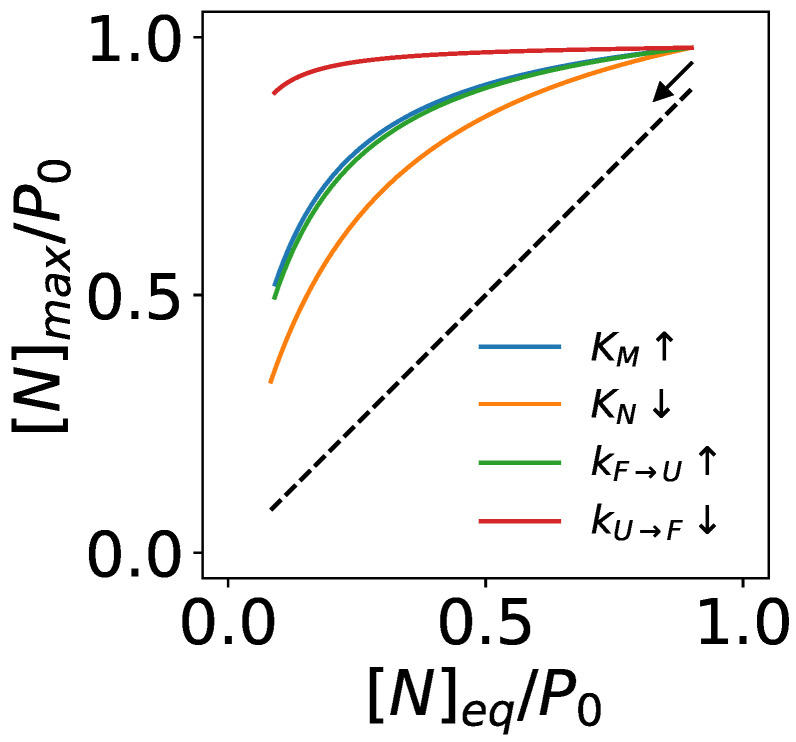
The capacity of ATP-driven chaperones to maintain elevated native fractions in response to destabilizing mutations in a protein substrate. The kinetic parameters of the wild-type protein are $K_{M}=10^{2}$, $K_{F}=10$, $K_{N}=10^{2}$, $k_{U\to F}=0.1\;s^{-1}$, and $k_{a}=10^{6}\;M^{-1}\cdot s^{-1}$; the concentration of the open chaperone is set to [O]=1 µM.

**Figure 6 biomolecules-12-00832-f006:**
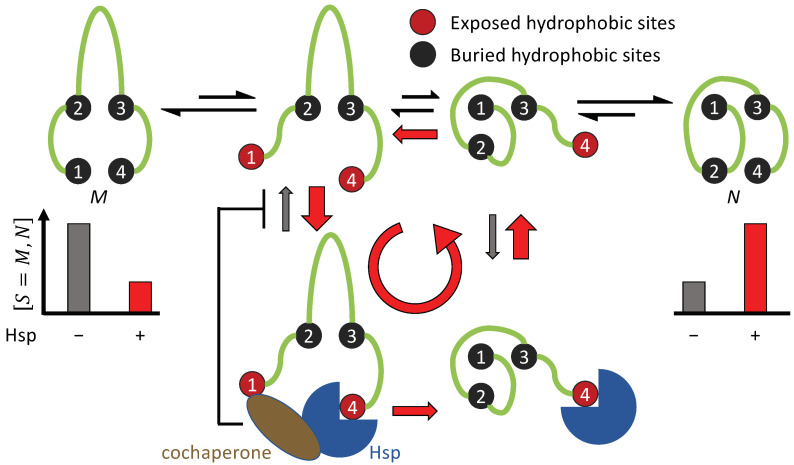
An ATP-driven chaperone (Hsp) may favor the protein folding pathway that exposes few cochaperone binding sites and drive the protein to a different conformation (*N*) than its most stable conformation (*M*) in the absence of the chaperone.

**Table 1 biomolecules-12-00832-t001:** The reactions in chaperone-mediated protein folding. These reactions are depicted in Figure 1. ATP hydrolysis and nucleotide exchange occur and inject chemical energy in the chaperone cycle.

$U\overset{k_{U\to M}}{\underset{k_{M\to U}}{⇌}}M$	Misfolding and aggregation; $K_{M}=k_{U\to M}/k_{M\to U}$
$U\overset{k_{U\to F}}{\underset{k_{F\to U}}{⇌}}F$	Transition between intermediate conformations; $K_{F}=k_{U\to F}/k_{F\to U}$
$F\overset{k_{F\to N}}{\underset{k_{N\to F}}{⇌}}N$	Folding to native structure; $K_{N}=k_{F\to N}/k_{N\to F}$
$S+O\overset{k_{a,S}}{\underset{k_{d,S}}{⇌}}SO$	Substrate in S=U,F conformations binding to the open chaperone
$SO\overset{k_{S,O\to C}}{\underset{k_{S,C\to O}}{⇌}}SC$	Transition of chaperone between open and closed states
$UH\overset{k_{U\to F,H}}{\underset{k_{F\to U,H}}{⇌}}FH$	Conversion of protein bound to chaperone in H=C,O states

**Table 2 biomolecules-12-00832-t002:** A hypothetical set of equilibrium constants that are consistent with the measured Cdc37 affinities of the wild-type B-raf and the V600E mutant. The dissociation constants are similar between the inactive wild-type and the active mutant.

	$K_{M}$	$K_{F}$	$K_{N}$	$K_{D}^{*}$ (μM)	${\left[N\right]}_{\mathrm{eq}}/P_{0}$	$K_{D}$ (μM)
wild-type	100	0.1	80	0.0092	0.07	1.0
V600E	10.24	0.4	80	0.0092	0.73	0.4

## Data Availability

Not applicable.

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
