# Peer review of "Non-Equilibrium Protein Folding and Activation by ATP-Driven Chaperones"

_biomolecules, 2022, doi:10.3390/biom12060832_

Round 1
Reviewer 1 Report
In this article, the author sets out to understand principles behind ATP-driven chaperone folding. Overall the study is scientifically sound and provides novel insight into these complex mechanisms. The figures are nicely drawn and are easy to understand. Minor comments: There is now increasing evidence that both Hsp90 and Hsp70 are regulated through post-translational modification. We would ask the authors to mention this in the article and reference the following reviews "Pubmed ID: 32518165 and 32527727".
Author Response
Per the reviewer’s helpful suggestion, I have added a new paragraph in the Discussion to briefly address the regulation of the chaperones—including by post-translational modifications (citing the suggested references)—and its effects on non-equilibrium protein folding.
Reviewer 2 Report
It is regarded that ATP-driven chaperones stabilize out of thermal equilibrium the structures of client proteins, whereas the mechanism for non-equilibrium protein folding is still poorly understood. In this article, Huafeng Xu proposes a unifying principle that underlies the conversion of chemical energy from ATP hydrolysis to the conformational free energy associated with protein folding and activation. The author uses as a model the Hsp90-Hsp70-Hsp40 molecular chaperone complex. His theoretical results predict that some proteins may fold into thermodynamically unstable native structures with the assistance of ATP-driven chaperones, and that the native structures of some chaperone-dependent proteins may be shaped by their chaperone-mediated folding pathways.
The article is very interesting and quite challenging since it disputes (although does not contradict) some basic principles of the Anfinsen’s dogma. The logic applied by the author is good, the mathematical model is correct, and therefore, his conclusions are rational. Perhaps the main criticism is the fact that these postulates should be demonstrated experimentally. The model is 100% theoretical, but useful to feed the market of ideas.
Author Response
I thank the reviewer for the encouraging comments. Although this work is theoretical, I did include suggestions of experiments to test the theoretical predictions. As the reviewer suggested, maybe the ideas put forward in this manuscript will beget new experiments.